# Mirror Diffusion Models for Constrained and Watermarked Generation

**Guan-Horng Liu, Tianrong Chen, Evangelos A. Theodorou[†], Molei Tao[†]**
Georgia Institute of Technology, USA
{ghliu, tianrong.chen, evangelos.theodorou, mtao}@gatech.edu

## Abstract

Modern successes of diffusion models in learning complex, high-dimensional data distributions are attributed, in part, to their capability to construct diffusion processes with analytic transition kernels and score functions. The tractability results in a simulation-free framework with stable regression losses, from which reversed, generative processes can be learned at scale. However, when data is confined to a constrained set as opposed to a standard Euclidean space, these desirable characteristics appear to be lost based on prior attempts. In this work, we propose **Mirror Diffusion Models (MDM)**, a new class of diffusion models that generate data on convex constrained sets without losing any tractability. This is achieved by learning diffusion processes in a dual space constructed from a mirror map, which, crucially, is a standard Euclidean space. We derive efficient computation of mirror maps for popular constrained sets, such as simplices and $\ell_2$-balls, showing significantly improved performance of MDM over existing methods. For safety and privacy purposes, we also explore constrained sets as a new mechanism to embed invisible but quantitative information (*i.e.,* watermarks) in generated data, for which MDM serves as a compelling approach. Our work brings new algorithmic opportunities for learning tractable diffusion on complex domains.

## 1 Introduction

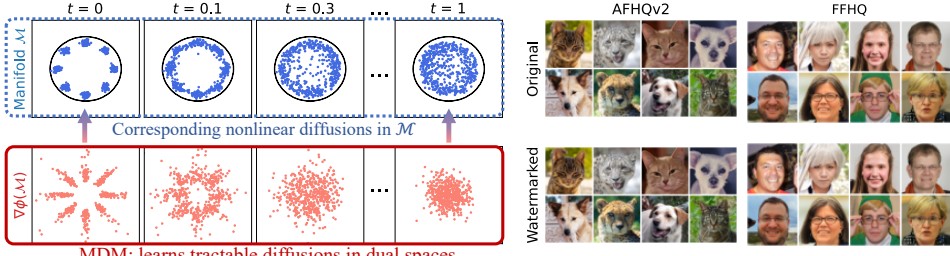

Figure 1: **Mirror Diffusion Models (MDM)** is a new class of diffusion models for convex constrained manifolds $\mathcal{M} \subseteq \mathbb{R}^d$. (**left**) Instead of learning score-approximate diffusions on $\mathcal{M}$, MDM applies a *mirror* map $\nabla\phi$ and learns *tractable* diffusions in its *unconstrained dual-space* $\nabla\phi(\mathcal{M}) = \mathbb{R}^d$. (**right**) We also present MDM for watermarked generation, where generated contents (*e.g.,* images) live in a high-dimensional *token* constrained set $\mathcal{M}$ that is certifiable only from the private user.

Diffusion models [1–3] have emerged as powerful generative models with their remarkable successes in synthesizing high-fidelity data such as images [4, 5], audio [6, 7], and 3D geometry [8, 9]. These models work by progressively diffusing data to noise, and learning the score functions (often

---

[†]Equal advising.

37th Conference on Neural Information Processing Systems (NeurIPS 2023).

Table 1: Comparison of different diffusion models for constrained generation on $\mathcal{M} \subseteq \mathbb{R}^d$. Rather than learning reflected diffusions on $\mathcal{M}$ with approximate scores, our Mirror Diffusion constructs a mirror map $\nabla\phi$ and lift the diffusion processes to the *unconstrained dual space* $\nabla\phi(\mathcal{M}) = \mathbb{R}^d$, inheriting favorable features from standard Euclidean-space diffusion models. Constraints are satisfied by construction via the inverse map $\nabla\phi^*$; that is, $\nabla\phi^*(y) \in \mathcal{M}$ for all $y \in \nabla\phi(\mathcal{M})$.

| Diffusion models | Domain | Tractable conditional score | Simulation-free training | Regression objective |
|---|---|---|---|---|
| *Reflected Diffusion* | | | | |
| Fishman et al. [18] | $\mathcal{M}$ | ✗ | ✗ | ✗ |
| Lou and Ermon [25] | $\mathcal{M}$ | ✗ | $\triangle^2$ | ✓ |
| *Mirror Diffusion* | | | | |
| This work (ours) | $\nabla\phi(\mathcal{M})$ | ✓ | ✓ | ✓ |

parameterized by neural networks) to reverse the processes [10]; the reversed processes thereby provide transport maps that generate data from noise. Modern diffusion models [11–13] often employ diffusion processes whose transition kernels are analytically available. This characteristic enables *simulation-free* training, bypassing the necessity to simulate the underlying diffusion processes by directly sampling the diffused data. It also leads to tractable conditional score functions and simple regression objectives, facilitating computational scalability for high-dimensional problems. In standard Euclidean spaces, tractability can be accomplished by designing the diffusion processes as linear stochastic differential equations, but doing so for non-Euclidean spaces is nontrivial.

Recent progress of diffusion models has expanded their application to non-Euclidean spaces, such as Riemannian manifolds [14, 15], where data live in a curved geometry. The development of generative models on manifolds has enabled various new applications, such as modeling earth and climate events [16] and learning densities on meshes [17]. In this work, we focus on generative modeling on *constrained sets* (also called *constrained manifolds* [18]) that are defined by a set of inequality constraints. Such constrained sets, denoted by $\mathcal{M} \subseteq \mathbb{R}^d$, are also ubiquitous in several scientific fields such as computational statistics [19], biology [20], and robotics [21, 22].

Previous endeavors in the direction of constrained generation have primarily culminated in reflected diffusions [23, 24], which reflect the direction at the boundary $\partial\mathcal{M}$ to ensure that samples remain inside the constraints. Unfortunately, reflected diffusions do not possess closed-form transition kernels [18, 25], thus necessitating the approximation of the conditional score functions that can hinder learning performance. Although simulation-free training is still possible[2], it comes with a computational overhead due to the use of geometric techniques [25]. From a theoretical standpoint, reflected Langevin dynamics are widely regarded for their extended mixing time [26, 27]. This raises practical concerns when simulating reflected diffusions, as prior methods often adopted special treatments such as thresholding and discretization [18, 25].

An alternative diffusion for constrained sampling is the Mirror Langevin Dynamics (MLD [28]). MLD, as a subclass of Riemannian Langevin Dynamics [29, 30], is tailored specifically to convex constrained sampling. Specifically, MLD constructs a strictly convex function $\phi$ so that its gradient map, often referred to as the *mirror map* $\nabla\phi : \mathcal{M} \to \mathbb{R}^d$, defines a nonlinear mapping from the initial constrained set to an *unconstrained dual space*. Despite extensive investigation of MLD [31–34], its potential as a foundation for designing diffusion generative models remains to be comprehensively examined. Thus, there is a need to explore the potential of MLD for designing new, effective, and tailored diffusion models for constrained generation.

With this in mind, we propose **Mirror Diffusion Models (MDM)**, a new class of diffusion generative models for convex constrained sets. Similar to MLD, MDM utilizes mirror maps $\nabla\phi$ that transform data distributions on $\mathcal{M}$ to its dual space $\nabla\phi(\mathcal{M})$. From there, MDM learns a *dual-space diffusion* between the mirrored distribution and Gaussian prior in the dual space. Note that this is in contrast to MLD, which constructs invariant distributions on the initial constrained set; MDM instead considers dual-space priors. Since the dual space is also unconstrained Euclidean space $\nabla\phi(\mathcal{M}) = \mathbb{R}^d$, MDM

---

[2]Lou and Ermon [25] proposed a semi-simulation-free training with the aid of geometric techniques.

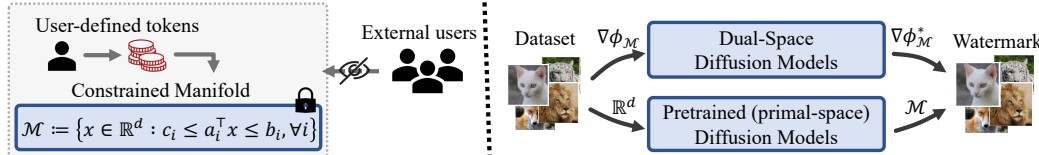

Figure 2: MDM for watermarked generation: (**left**) We first construct a constrained set $\mathcal{M}$ based on a set of user-defined tokens private to other users. (**right**) MDM can be instantiated by either learning the corresponding dual-space diffusions, or projecting pretrained, *i.e.,* unwatermarked, diffusion models onto $\mathcal{M}$. In both cases, MDM embeds watermarks that are certifiable only from the user.

preserves many important features of standard diffusion models, such as simulation-free training and tractable conditional scores, which yields simple regression objectives; see Table 1. After learning the dual-space distribution, MDM applies the inverse mapping $\nabla \phi^*$, transforming mirrored samples back to the constrained set. We propose efficient construction of mirror maps and demonstrate that MDM outperforms prior reflected diffusion models on common constrained sets, such as $\ell_2$-balls and simplices.

Moreover, we show that MDM also stands as a novel approach for *watermarked generation*, a technique that aims to embed undetectable information (*i.e.,* watermarks) into generated contents to prevent unethical usage and ensure copyright protection. This newly emerging goal nevertheless attracted significant attention in natural language generation [35–37], while what we will present is a general algorithmic strategy demonstrated for image generations.

To generate watermarked contents that are private to individual users, we begin by constructing a constrained set $\mathcal{M}$ based on user-defined tokens (see Figure 2). MDM can then be instantiated to learn the dual-space distribution. Alternatively, MDM can distill pretrained diffusion models [38] by projecting the generated data onto the constraint set. In either case, MDM generates watermarked contents that can be certified only by users who know the tokens. Our empirical results suggest that MDM serves as a competitive approach to prior watermarking techniques for diffusion models, such as those proposed by Zhao et al. [39], by achieving lower (hence better) FID values.

In summary, we present the following contributions:

- We introduce **Mirror Diffusion Models (MDM)**, a new class of diffusion models that utilizes mirror maps to enable constrained generation with Euclidean-space diffusion models.

- We propose efficient computation of mirror maps on common constrained sets, including balls and simplices. Our results demonstrate that MDM consistently outperforms previous reflected diffusion models.

- As a novel application of constrained generation, we explore MDM's potential for watermarking and copyrighting diffusion models where constraints serve as private tokens. We show that it yields competitive performance compared to prior methods.

## 2 Preliminary on Euclidean-space Diffusion Models

**Notation** $\mathcal{M} \subseteq \mathbb{R}^d$ denotes a convex constrained set. We preserve $x \in \mathcal{M}$ and $y \in \mathbb{R}^d$ to better distinguish variables on constrained sets and standard $d$-dimensional Euclidean space. $\boldsymbol{I} \in \mathbb{R}^{d \times d}$ denotes the identity matrix. $x_i$ denotes the $i$-th element of a vector $x$. Similarly, $[A]_{ij}$ denotes the the element of a matrix $A$ at $i$-th row and $j$-th column.

**Diffusion process** Given $y_0 \in \mathbb{R}^d$ sampled from some Euclidean-space distribution, conventional diffusion models define the diffusion process as a *forward* Markov chain with the joint distribution:

$$q(y_{1:T}|y_0) = \prod_{t=1}^{T} q(y_t|y_{t-1}), \quad q(y_t|y_{t-1}) := \mathcal{N}(y_t; \sqrt{1-\beta_t}y_{t-1}, \beta_t \boldsymbol{I}), \tag{1}$$

which progressively injects Gaussian noise to $y_0$ such that, for a sufficiently large $T$ and a properly chosen noise schedule $\beta_t \in (0, 1)$, $y_T$ approaches an approximate Gaussian, *i.e.,* $q(y_T) \approx \mathcal{N}(0, \boldsymbol{I})$.

Equation (1) has tractable marginal densities, and two of which that are of particular interest are:

$$q(y_t|y_0) = \mathcal{N}(y_t; \sqrt{\bar{\alpha}_t}y_0, (1-\alpha_t)\boldsymbol{I}), \quad q(y_{t-1}|y_t, y_0) = \mathcal{N}(y_{t-1}; \tilde{\mu}_t(y_t, y_0), \tilde{\beta}_t\boldsymbol{I}), \quad (2)$$

$$\tilde{\mu}_t(y_t, y_0) := \frac{\sqrt{\bar{\alpha}_{t-1}}\beta_t}{1-\bar{\alpha}_t}y_0 + \frac{\sqrt{1-\beta_t}(1-\bar{\alpha}_{t-1})}{1-\bar{\alpha}_t}y_t, \quad \tilde{\beta}_t := \frac{1-\bar{\alpha}_{t-1}}{1-\bar{\alpha}_t}\beta_t, \quad \bar{\alpha}_t := \prod_{s=0}^{t}(1-\beta_s).$$

The tractable marginal $q(y_t|y_0)$ enables direct sampling of $y_t$ without simulating the forward Markov chain (1), *i.e.,* $y_t|y_0$ can be sampled in a *simulation-free* manner. It also suggests a closed-form score function $\nabla \log q(y_t|y_0)$. The marginal $q(y_{t-1}|y_t, y_0)$ hints the optimal reverse process given $y_0$.

**Generative process**    The generative process, which aims to *reverse* the forward diffusion process (1), is defined by a *backward* Markov chain with the joint distribution:

$$p_\theta(y_{0:T}) = p_T(y_T)\prod_{t=1}^{T}p_\theta(y_{t-1}|y_t), \quad p_\theta(y_{t-1}|y_t) := \mathcal{N}(y_{t-1}; \mu_\theta(y_t, t), \tilde{\beta}_t\boldsymbol{I}). \quad (3)$$

Here, the mean $\mu_\theta$ is typically parameterized by some DNNs, *e.g.,* U-net [40], with $\theta$. Equations (1) and (3) imply an evidence lower bound (ELBO) for maximum likelihood training, indicating that the optimal prediction of $\mu_\theta(y_t, t)$ should match $\tilde{\mu}_t(y_t, y_0)$ in Equation (2).

**Parameterization and training**    There exists many different ways to parameterize $\mu_\theta(y_t, t)$, and each of them leads to a distinct training objective. For instance, Ho et al. [2] found that

$$\mu_\theta(y_t, t) := \frac{1}{\sqrt{1-\beta_t}}\Big(y_t - \frac{\beta_t}{\sqrt{1-\bar{\alpha}_t}}\epsilon_\theta(y_t, t)\Big), \quad (4)$$

achieves better empirical results compared to directly predicting $y_0$ or $\tilde{\mu}_t$. This parameterization aims to predict the "noise" injected to $y_t \sim q(y_t|y_0)$, as the ELBO reduces to a simple regression:

$$\mathcal{L}(\theta) := \mathbb{E}_{t,y_0,\epsilon}\left[\lambda(t)\|\epsilon - \epsilon_\theta(y_t, t)\|\right]. \quad (5)$$

where $y_t = \sqrt{\bar{\alpha}_t}y_0 + \sqrt{1-\alpha_t}\epsilon$, $\epsilon \sim \mathcal{N}(0, \boldsymbol{I})$, and $t$ sampled uniformly from $\{1, \cdots, T\}$. The weighting $\lambda(t)$ is often set to 1. Once $\epsilon_\theta$ is properly trained, we can generate samples in Euclidean space by simulating Equation (3) backward from $y_T \sim \mathcal{N}(0, \boldsymbol{I})$.

# 3   Mirror Diffusion Models (MDM)

In this section, we present a novel class of diffusion models, Mirror Diffusion Models, which are designed to model probability distributions $p_{\text{data}}(x)$ supported on a convex constrained set $\mathcal{M} \subseteq \mathbb{R}^d$ while retraining the computational advantages of Euclidean-space diffusion models.

## 3.1   Dual-Space Diffusion

**Mirror map**    Following the terminology in Li et al. [34], let $\phi : \mathcal{M} \to \mathbb{R}$ be a twice differentiable[3] function that is strictly convex and satisfying that $\lim_{x\to\partial\mathcal{M}}\|\nabla\phi(x)\| \to \infty$ and $\nabla\phi(\mathcal{M}) = \mathbb{R}^d$. We call its gradient map $\nabla\phi : \mathcal{M} \to \mathbb{R}^d$ the *mirror map* and its image, $\nabla\phi(\mathcal{M}) = \mathbb{R}^d$, as its *dual/mirror space*. Let $\phi^* : \mathbb{R}^d \to \mathbb{R}$ be the dual function of $\phi$ defined by

$$\phi^*(y) = \sup_{x\in\mathcal{M}}\langle x, y\rangle - \phi(x) \quad (6)$$

A notable property of this dual function $\phi^*(y)$ is that its gradient map *reverses* the mirror map. In other words, we have $\nabla\phi^* = (\nabla\phi)^{-1}$, implying that

$$\nabla\phi^*(\nabla\phi(x)) = x \text{ for all } x \in \mathcal{M}, \quad \nabla\phi(\nabla\phi^*(y)) = y \text{ for all } y \in \nabla\phi(\mathcal{M}) = \mathbb{R}^d.$$

The functions $(\nabla\phi, \nabla\phi^*)$ define a nonlinear, yet bijective, mapping between the convex constrained set $\mathcal{M}$ and Euclidean space $\mathbb{R}^d$. As a result, MDM does not require building diffusion models on $\mathcal{M}$, as done in previous approaches, but rather they learn a standard Euclidean-space diffusion model in the corresponding dual space $\nabla\phi(\mathcal{M}) = \mathbb{R}^d$.

---

[3]We note that standard mirror maps require twice differentiability to induce a (Riemannian) metric on the mirror space for constructing MLD. For our MDM, however, twice differentiability is needed only for Equation (8), and continuous differentiability, *i.e., $C^1$*, would suffice for training.

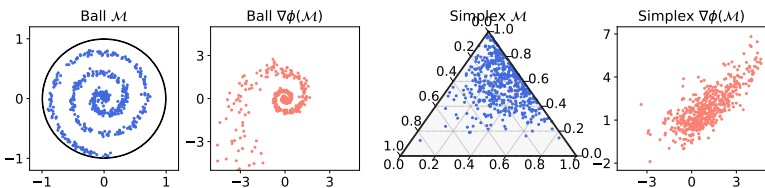

Figure 3: Examples of how mirror maps $\nabla\phi$ pushforward constrained distributions to unconstrained ones, for (**left**) an $\ell_2$-ball where $\mathcal{M} := \{x \in \mathbb{R}^2 : \|x\|_2^2 < R\}$ and (**right**) a simplex $\Delta_3$ where $\mathcal{M} := \{x \in \mathbb{R}^2 : \sum_{i=1}^2 x_i \leq 1, x_i \geq 0\}$. Note that for simplex, $x_3 = 1 - x_1 - x_2$ is a redundant coordinate; see Section 3.2 for more details.

Essentially, the goal of MDM is to model the *dual-space distribution*:

$$\tilde{p}_{\text{data}}(y) := ([\nabla\phi_\sharp]p_{\text{data}})(x), \tag{7}$$

where $\sharp$ is the push-forward operation and $y = \nabla\phi(x)$ is the mirror of data $x \in \mathcal{M}$.

**Training and generation**  MDM follows the same training procedure of Euclidean-space diffusion models, except with an additional push-forward given by Equation (7). Similarly, its generation procedure includes an additional step that maps the dual-space samples $y_0$, generated from Equation (3), back to the constrained set via $\nabla\phi^*(y_0) \in \mathcal{M}$.

**Tractable variational bound**  Given a datapoint $x \in \mathcal{M}$, its mirror $y_0 = \nabla\phi(x) \in \mathbb{R}^d$, and the dual-space Markov chain $\{y_1, \cdots, y_T\}$, the ELBO of MDM can be computed by

$$L_{\text{ELBO}}(x) := \log|\det\nabla^2\phi^*(y_0)| + \tilde{L}_{\text{ELBO}}(y_0), \tag{8}$$

where we apply the change of variables theorem [41] w.r.t. the mapping $x = \nabla\phi^*(y_0)$. The dual-space ELBO $\tilde{L}_{\text{ELBO}}(y_0)$ shares the same formula with the Euclidean-space ELBO [42, 43],

$$\tilde{L}_{\text{ELBO}}(y_0) := D_{\text{KL}}(q(y_T|y_0) \,||\, p(y_T)) + \sum_{t=1}^{T-1} D_{\text{KL}}(q(y_t|y_{t+1}, y_0) \,||\, p_\theta(y_t|y_{t+1})) - \log p_\theta(y_0|y_1). \tag{9}$$

It should be noted that (8,9) provide a *tractable* variational bound for constrained generation. This stands in contrast to concurrent methods relying on reflected diffusions, whose ELBO entails either intractable [18] or approximate [25] scores that could be restricted when dealing with more complex constrained sets. While, in practice, MDM is trained using the regression objective in Equation (5), computing tractable ELBO might be valuable for independent purposes such as likelihood estimation.

### 3.2  Efficient Computation of Mirror Maps

What remains to be answered pertains to the construction of mirror maps for common constraint sets such as $\ell_2$-balls, simplices, polytopes, and their products. In particular, we seek efficient, preferably closed-form, computation for $\nabla\phi$, $\nabla\phi^*$ and $\nabla^2\phi^*$.

$\ell_2$-**Ball**  Let $\mathcal{M} := \{x \in \mathbb{R}^d : \|x\|_2^2 < R\}$ denote the $\ell_2$-ball of radius $R$ in $\mathbb{R}^d$. We consider the common log-barrier function:

$$\phi_{\text{ball}}(x) := -\gamma\log(R - \|x\|_2^2). \tag{10}$$

where $\gamma \in \mathbb{R}^+$. The mirror map, its inverse, and the Hessian can be computed analytically by

$$\nabla\phi_{\text{ball}}(x) = \frac{2\gamma x}{R - \|x\|_2^2}, \qquad \nabla\phi_{\text{ball}}^*(y) = \frac{Ry}{\sqrt{R\|y\|_2^2 + \gamma^2} + \gamma},$$

$$\nabla^2\phi_{\text{ball}}^*(y) = \frac{R}{R\sqrt{\|y\|_2^2 + \gamma^2} + \gamma}\left(\boldsymbol{I} - \frac{R}{\left(R\sqrt{\|y\|_2^2 + \gamma^2} + \gamma\right)\sqrt{R\|y\|_2^2 + \gamma^2}}yy^\top\right). \tag{11}$$

We refer to Appendix A for detailed derivation. Figure 3 visualizes such a mirror map for an $\ell_2$-ball in 2D. It is worth noting that while the mirror map of log-barriers generally does not admit an analytical inverse, this is not the case for the $\ell_2$-ball constraints.

**Simplex**   Given a $(d+1)$-dimensional simplex, $\Delta_{d+1} := \{x \in \mathbb{R}^{d+1} : \sum_{i=1}^{d+1} x_i = 1, x_i \geq 0\}$, we follow standard practices [18, 25] by constructing the constrained set $\mathcal{M} := \{x \in \mathbb{R}^d : \sum_{i=1}^{d} x_i \leq 1, x_i \geq 0\}$ and define $x_{d+1} := 1 - \sum_{i=1}^{d} x_i$.[4]

While conventional log-barriers may remain a viable option, a preferred alternative that is widely embraced in the MLD literature for enforcing simplices [28, 44] is the entropic function [45, 46]:

$$\phi_{\text{simplex}}(x) := \sum_{i=1}^{d} x_i \log(x_i) + \left(1 - \sum_{i=1}^{d} x_i\right) \log\left(1 - \sum_{i=1}^{d} x_i\right). \tag{12}$$

The mirror map, its inverse, and the hessian of the dual function can be computed analytically by

$$[\nabla \phi_{\text{simplex}}(x)]_i = \log x_i - \log\left(1 - \sum_{i=1}^{d} x_i\right), \qquad [\nabla \phi_{\text{simplex}}^*(y)]_i = \frac{e^{y_i}}{1 + \sum_{i=1}^{d} e^{y_i}},$$

$$[\nabla^2 \phi_{\text{simplex}}^*(y)]_{ij} = \frac{e^{y_i}}{1 + \sum_{i=1}^{d} e^{y_i}} \mathbb{I}(i = j) - \frac{e^{y_i} e^{y_j}}{\left(1 + \sum_{i=1}^{d} e^{y_i}\right)^2}, \tag{13}$$

where $\mathbb{I}(\cdot)$ is an indicator function. Again, we leave the derivation of Equation (13) to Appendix A. Figure 3 visualizes the entropic mirror map for a 3-dimensional simplex $\Delta_3$.

**Polytope**   Let the constrained set be $\mathcal{M} := \{x \in \mathbb{R}^d : c_i < a_i^\top x < b_i, \forall i \in \{1, \cdots, m\}\}$, where $c_i, b_i \in \mathbb{R}$ and $a_i \in \mathbb{R}^d$ are linearly independent to each other. We consider the standard log-barrier:

$$\phi_{\text{polytope}}(x) := -\sum_{i=1}^{m} \left(\log\left(\langle a_i, x\rangle - c_i\right) + \log\left(b_i - \langle a_i, x\rangle\right)\right) + \sum_{j=m+1}^{d} \frac{1}{2}\langle a_j, x\rangle^2, \tag{14}$$

$$\nabla \phi_{\text{polytope}}(x) = \sum_{i=1}^{m} s_i(\langle a_i, x\rangle)a_i + \sum_{j=m+1}^{d} \langle a_j, x\rangle a_j, \tag{15}$$

where the monotonic function $s_i : (c_i, b_i) \to \mathbb{R}$ is given by $s_i = \frac{-1}{\langle a_i, x\rangle - c_i} + \frac{-1}{\langle a_i, x\rangle - b_i}$, and $\{a_j\}$ do not impose any constraints and can be chosen arbitrarily as long as they span $\mathbb{R}^d$ with $\{a_i\}$ (see below for more details); hence $\nabla \phi_{\text{polytope}}(\mathcal{M}) = \mathbb{R}^d$. While its inverse, $\nabla \phi_{\text{polytope}}^*$, does not admit closed-form in general, it does when all $d$ constraints are *orthonormal*:

$$\nabla \phi_{\text{polytope}}^*(y) = \sum_{i=1}^{m} s_i^{-1}(\langle a_i, y\rangle)a_i + \sum_{j=m+1}^{d} \langle a_j, y\rangle a_j. \tag{16}$$

Figure 4: Comparison between $s_i$ induced by standard log-barriers *vs.* hyperbolic tangents in Eq. (18).

Here, $s_i^{-1} : \mathbb{R} \to (c_i, b_i)$ is the inverse of $s_i$. Essentially, (15,16) manipulate the coefficients of the bases from which the polytope set is constructed. These manipulations are nonlinear yet bijective, defined uniquely by $s_i$ and $s_i^{-1}$, for each orthonormal basis $a_i$ or, equivalently, for each constraint.

Naively implementing the mirror map via (15,16) can be problematic due to *(i)* numerical instability of $s_i$ at boundaries (see Figure 4), *(ii)* the lack of closed-form solution for its inverse $s_i^{-1}$, and *(iii)* an *one-time* computation of orthonormal bases, which scales as $\mathcal{O}(d^3)$ for orthonormalization methods such as Gram-Schmidt, Householder or Givens [e.g., 47]. Below, we devise a modification for efficient and numerically stable computation, which will be later adopted for watermarked generation.

**Improved computation for Polytope (15,16)**   Given the fact that only the first $m$ orthonormal bases $\{a_1, \cdots, a_m\}$ matter in the computation of (15,16), as the remaining $(d - m)$ orthonormal bases merely involve orthonormal projections, we can simplify the computational (15,16) to

$$\nabla \phi_{\text{poly}}(x) = x + \sum_{i=1}^{m} \left(s_i(\langle a_i, x\rangle) - \langle a_i, x\rangle\right) a_i, \quad \nabla \phi_{\text{poly}}^*(y) = y + \sum_{i=1}^{m} \left(s_i^{-1}(\langle a_i, y\rangle) - \langle a_i, y\rangle\right) a_i, \tag{17}$$

---

[4]The transformation allows $x$ to be bounded in a constrained set $\mathcal{M} \subseteq \mathbb{R}^d$ rather than a hyperplane in $\mathbb{R}^{d+1}$.

which, intuitively, add and subtract the coefficients of the first $m$ orthonormal bases while leaving the rest intact. This reduces the complexity of computing orthonormal bases to $\mathcal{O}(md^2) \approx \mathcal{O}(d^2)$ and improves the numerical accuracy of inner-product multiplication, which can be essential for high-dimensional applications when $d$ is large.

To address the instability of $s_i$ and intractability of $s_i^{-1}$ for log-barriers, we re-interpret the mirror maps (15,16) as the changes of coefficient bases. As this implies that any nonlinear bijective mapping with a tractable inverse would suffice, we propose a rescaling of the hyperbolic tangent:

$$s_i(\langle a_i, x\rangle) := \left(\tanh^{-1} \circ \text{rescale}_i\right)(\langle a_i, x\rangle), \quad s_i^{-1}(\langle a_i, y\rangle) := \left(\text{rescale}_i^{-1} \circ \tanh\right)(\langle a_i, y\rangle), \quad (18)$$

where $\text{rescale}_i(z) := 2\frac{z-c_i}{(b_i-c_i)} - 1$ rescales the range from $(c_i, b_i)$ to $(-1, 1)$. It is clear from Figure 4 that, compared to the $s_i$ induced by the log-barrier, the mapping (18) is numerical stable at the boundaries and admits tractable inverse.

**Complexity**    Table 2 summarizes the complexity of the mirror maps for each constrained set. We note that all computations are parallelizable, inducing nearly no computational overhead.

Table 2: Complexity of $\nabla\phi$ and $\nabla\phi^*$ for each constrained set.

| $\ell_2$-Ball | Simplex | Polytope |
|---|---|---|
| $\mathcal{O}(d)$ | $\mathcal{O}(d)$ | $\mathcal{O}(md)$ |

**Remark 1.** *Equations* (17) *and* (18) *can be generalized to non-orthonormal constraints by adopting* $\langle \tilde{a}_i, x\rangle$ *and* $\langle \tilde{a}_i, y\rangle$, *where* $\tilde{a}_i$ *is the i-th row of the matrix* $\tilde{A} := (A^\top A)^{-1} A^\top$ *and* $A := [a_1, \cdots, a_m]$. *Indeed, when* $A$ *is orthonormal, we recover* $\tilde{a}_i = a_i$. *We leave more discussions to Appendix B.*

## 4    Related Work

**Constrained sampling**    Sampling from a probability distribution on a constrained set has been a long-standing problem not only due to its extensive applications in, *e.g.,* topic modeling [48, 49] and Bayesian inference [50, 51], but its unique challenges in non-asymptotic analysis and algorithmic design [28, 32–34, 52, 53]. Mirror Langevin is a special instance of endowing Langevin dynamics with a Riemannian metric [30, 54], and it chooses a specific reshaped geometry so that dynamics have to travel infinite distance in order to cross the boundary of the constrained set. It is a sampling generalization of the celebrated Mirror Descent Algorithm [45] for convex constrained optimization, and the convexity of constrained set leads to the existence of a mirror map. However, Mirror Langevin is designed to draw samples from an unnormalized density supported on a constrained set. This stands in contrast to our MDM, which instead tackles problems of constrained *generation*, where only samples, not unnormalized density, are available, sharing much similarity to conventional generative modeling [3, 55]. In addition, Mirror Langevin can be understood to go back and forth between primal (constrained) and mirror (unconstrained) spaces, while MDM only does one such round trip.

**Diffusion Models in non-Euclidean spaces**    There has been growing interest in developing diffusion generative models that can operate on domains that are not limited to standard Euclidean spaces, *e.g.,* discrete domains [56, 57] and equality constraints [58]. Seminar works by De Bortoli et al. [14] and Huang et al. [15] have introduced diffusion models on Riemannian manifolds, which have shown promising results in, *e.g.,* biochemistry [59, 60] and rotational groups [61, 62]. Regarding diffusion models on constrained manifolds, most concurrent works consider similar convex constrained sets such as simplices and polytopes [18, 25, 63]. Our MDM works on the same classes of constraints and includes additionally $\ell_2$-ball, which is prevalent in social science domains such as opinion dynamics [64, 65]. It is also worth mentioning that our MDM may be conceptually similar to Simplex Diffusion (SD) [63] which reconstructs samples on simplices similar to the mirror mapping. However, SD is designed specifically for simplices and Dirichlet distributions, while MDM is applicable to a broader class of convex constrained sets. Additionally, SD adopts Cox-Ingersoll-Ross processes [66], yielding a framework that, unlike MDM, is not simulation-free.

**Latent Diffusion Models (LDMs)**    A key component to our MDM is the construction of mirror map, which allows us to define diffusion models in a different space than the original constrained set, thereby alleviating many computational difficulties. This makes MDM a particular type of LDMs [5, 67, 68], which generally consider structurally advantageous spaces for learning diffusion models. While LDMs typically *learn* such latent spaces either using pretrained model or on the fly using, *e.g.,* variational autoencoders, MDM instead derives analytically a latent (dual) space through a mirror map tailored specifically to the constrained set. Similar to LDM [69], the diffusion processes of MDM on the initial constrained set are implicitly defined and can be highly nonlinear.

Table 3: Results of $\ell_2$-**ball constrained sets** on five synthetic datasets of various dimension $d$. We report Sliced Wasserstein [70] w.r.t. 1000 samples, averaged over three trials, and include constraint violations for unconstrained diffusion models. Note that Reflected Diffusion [18] and MDM satisfy constraints by design. Our MDM clearly achieves similar or better performance to standard diffusion models while fully respecting the constraints. Other metrics, *e.g.*, $\mathcal{W}_1$, are reported in Appendix C. sc

| | $d = 2$ | $d = 2$ | $d = 6$ | $d = 8$ | $d=20$ |
|---|---|---|---|---|---|
| Sliced Wasserstein ↓ | | | | | |
| DDPM [2] | 0.0704 ± 0.0095 | 0.0236 ± 0.0048 | **0.0379** ± 0.0015 | **0.0231** ± 0.0020 | 0.0200 ± 0.0034 |
| Reflected [18] | 0.0642 ± 0.0181 | 0.0491 ± 0.0103 | 0.0609 ± 0.0055 | 0.0881 ± 0.0010 | 0.0574 ± 0.0065 |
| MDM (ours) | **0.0544** ± 0.0070 | **0.0214** ± 0.0025 | 0.0467 ± 0.0096 | 0.0292 ± 0.0017 | **0.0159** ± 0.0044 |
| Constraint violation (%) ↓ | | | | | |
| DDPM [2] | 0.00 ± 0.00 | 0.00 ± 0.00 | 8.67 ± 0.87 | 13.60 ± 0.62 | 19.33 ± 1.29 |

Table 4: Results of **simplices constrained sets** on five synthetic datasets of various dimension $d$.

| | $d = 3$ | $d = 3$ | $d = 7$ | $d = 9$ | $d=20$ |
|---|---|---|---|---|---|
| Sliced Wasserstein ↓ | | | | | |
| DDPM [2] | 0.0089 ± 0.0002 | **0.0110** ± 0.0032 | **0.0047** ± 0.0004 | 0.0053 ± 0.0003 | 0.0031 ± 0.0003 |
| Reflected [18] | 0.0233 ± 0.0019 | 0.0336 ± 0.0009 | 0.0411 ± 0.0039 | 0.0897 ± 0.0112 | 0.0231 ± 0.0011 |
| MDM (ours) | **0.0074** ± 0.0008 | 0.0169 ± 0.0016 | 0.0051 ± 0.0006 | **0.0040** ± 0.0003 | **0.0027** ± 0.0000 |
| Constraint violation (%) ↓ | | | | | |
| DDPM [2] | 0.73 ± 0.12 | 14.40 ± 1.39 | 11.63 ± 0.90 | 27.53 ± 0.57 | 68.83 ± 1.66 |

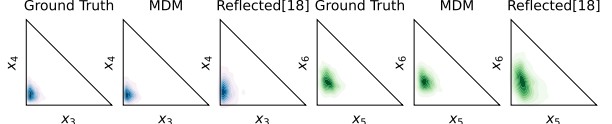

Figure 5: Comparison between MDM and Reflected Diffusion [18] in modeling a Dirichlet distribution on a 7-dimensional simplex. We visualize the joint densities between $x_{3:4}$ and $x_{5:6}$.

Table 5: Runtime and memory complexity w.r.t. DDPM [2].

| | Runtime | Memory |
|---|---|---|
| MDM | 108% | 100% |
| Reflected Diff. [18] | >1200% | 905% |

## 5 Experiment

### 5.1 Constrained Generation on Balls and Simplices

**Setup** We evaluate the performance of **Mirror Diffusion Model (MDM)** on common constrained generation problems, such as $\ell_2$-ball and simplex constrained sets with dimensions $d$ ranging from 2 to 20. Following Fishman et al. [18], we consider mixtures of Gaussian (Figure 1) and Spiral (Figure 3) for ball constraints and Dirichlet distributions [48] with various concentrations for simplex constraints. We compare MDM against standard unconstrained diffusion models, such as DDPM [2], and their constrained counterparts, such as Reflected Diffusion [18], using the same time-embedded fully-connected network and 1000 sampling time steps. Evaluation metrics include Sliced Wasserstein distance [70] and constraint violation. Other details are left to Appendix C.

**MDM surpasses Reflected Diffusion on all tasks.** Tables 3 and 4 summarize our quantitative results. For all constrained generation tasks, our MDM surpasses Reflected Diffusion by a large margin, with larger performance gaps as the dimension $d$ increases. Qualitatively, Figure 5 demonstrates how MDM better captures the underlying distribution than Reflected Diffusion. Additionally, Table 5 reports the relative complexity of both methods compared to simulation-free diffusion models such as DDPM. While Reflected Diffusion requires extensive computation due to the use of implicit score matching [71] and assertion of boundary reflection at every propagation step, our MDM constructs efficient, parallelizable, mirror maps in unconstrained dual spaces (see Section 3), thereby enjoying preferable simulation-free complexity and better numerical scalability.

**MDM matches DDPM without violating constraints.** It should be highlighted that, while DDPM remains comparable to MDM in terms of distributional metric, the generated samples often violate the designated constrained sets. As shown in Tables 3 and 4, these constraint violations worsen as the dimension increases. In contrast, MDM is by design violation-free; hence marrying the best of both.

Table 6: The 50k-FID for unconditional watermarked generation. *Watermark precision* denotes the percentage of constraint-satisfied images that are generated by MDMs, controlled by loosing/tightening the constrained sets. As MDM-dual is trained on constraint-projected images, we also report its *FIDs** w.r.t. the shifted distributions. The prior watermarked diffusion model [39] reported 5.03 on FFHQ and 4.32 on AFHQv2.

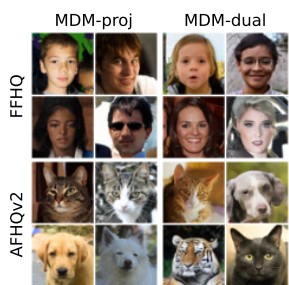

|  | FFHQ 64×64 | | | AFHQv2 64×64 | | |
|---|---|---|---|---|---|---|
| *Precision* | *59.3%* | *71.8%* | *93.3%* | *56.9%* | *75.0%* | *92.7%* |
| MDM-proj | 2.54 | 2.59 | 3.08 | 2.10 | 2.12 | 2.30 |
| MDM-dual | 2.96 | 4.57 | 15.74 | 2.23 | 2.86 | 6.79 |
|  | *2.65** | *2.93** | *4.40** | *2.21** | *2.32** | *3.05** |

Figure 6: Uncurated watermarked samples by MDMs. More samples can be found in Appendix C.

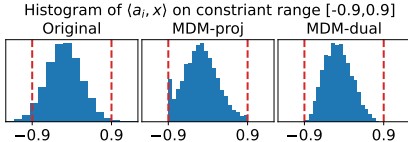

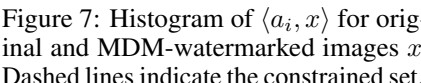

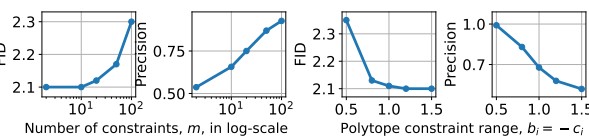

Figure 7: Histogram of $\langle a_i, x \rangle$ for original and MDM-watermarked images $x$. Dashed lines indicate the constrained set.

Figure 8: Ablation studies on how (**top**) number of constraints $m$ and (**bottom**) their ranges, *i.e.,* $(c_i, b_i)$, affect the FID and precision of MDM-proj on AFHQv2.

## 5.2 Watermarked Generation on High-Dimensional Image Datasets

**Setup** We present MDM as a new approach for watermarked generation, where the watermark is attached to samples within an orthonormal polytope $\mathcal{M} := \{x \in \mathbb{R}^d : c_i < a_i^\top x < b_i, \forall i\}$. These parameters $\{a_i, b_i, c_i\}_{i=1}^m$ serve as the *private tokens* visible only to individual users (see Figure 2). While our approach is application-agnostic, we demonstrate MDM mainly on image datasets, including both unconditional and conditional generation. Given a designated watermark precision, defined as the percentage of constraint-satisfied images that are generated by MDMs, we randomly construct tokens as $m$ orthonormalized Gaussian random vectors and instantiate MDMs by either projecting unwatermarked diffusion models onto $\mathcal{M}$ (**MDM-proj**), or learning the corresponding dual-space diffusions (**MDM-dual**). More precisely, MDM-proj projects samples generated by pretrained diffusion models to a constraint set whose parameters (*i.e.,* tokens) are visible only to the private user. In contrast, MDM-dual *learns* a dual-space diffusion model from the constrained-projected samples; hence, like other MDMs in Section 5.1, it is constraint-dependent. Other details are left to Appendix C.

**Unconditional watermark generation on FFHQ and AFHQv2** We first test out MMD on FFHQ [72] and AFHQv2 [73] on unconditional 64×64 image generation. Following Zhao et al. [39], we adopt EDM parameterization [38] and report in Table 6 the performance of MDM in generating watermarked images, quantified by the Frechet Inception Distance (FID) [74]. Compared to prior watermarked approaches, which require additional training of latent spaces [39], **MDM stands as a competitive approach for watermarked generation** by directly constructing analytic dual spaces from the tokens (*i.e.,* constraints) themselves. Intriguingly, despite the fact that MDM-dual learns a smoother dual-space distribution compared to the truncated one from MDM-proj (see Figure 7), the latter consistently achieves lower (hence better) FIDs. Since samples generated by MDM-dual still remain close to the watermarked distribution, as indicated in Table 6 by the low *FIDs** w.r.t. the shifted training statistics, we conjecture that the differences may be due to the significant distributional shift induced by naively projecting the training data onto the constraint set. This is validated, partly, in the ablation study from Figure 8, where we observe that both the FID and watermark precision improve as the number of constraints $m$ decreases, and as the constraint ranges $(c_i, b_i)$ loosen. We highlight this fundamental trade off between watermark protection and generation quality. Examples of watermarked images are shown in Figure 6 and Appendix C.

**Conditional generation on ImageNet256** Finally, we consider conditional watermarked generation on ImageNet 256×256. Specifically, we focus on image restoration tasks, where the goal

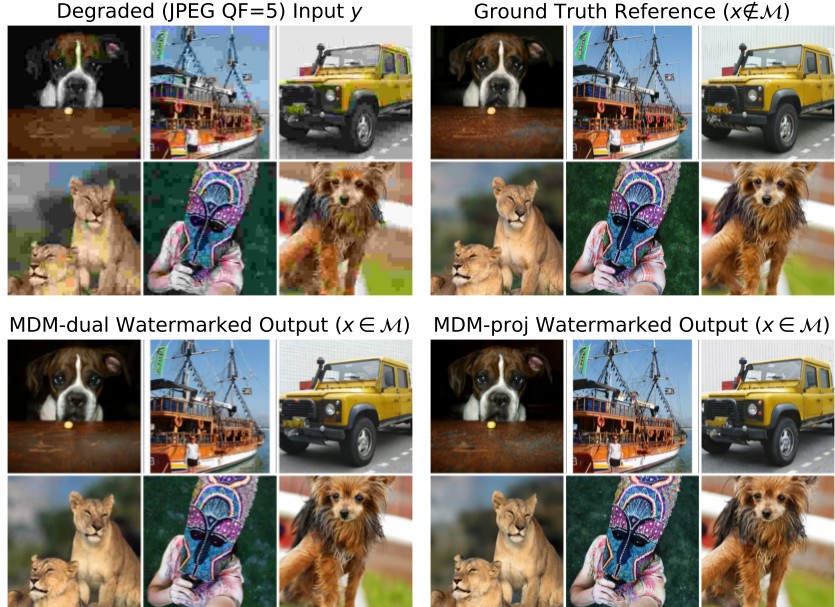

Degraded (JPEG QF=5) Input $y$      Ground Truth Reference ($x \notin \mathcal{M}$)

MDM-dual Watermarked Output ($x \in \mathcal{M}$)      MDM-proj Watermarked Output ($x \in \mathcal{M}$)

Figure 9: Conditional watermarked generation on ImageNet $256 \times 256$. Specifically, we consider the JPEG restoration task, where, given degraded, low-quality inputs $y$ (*upper-left*), we wish to generate their corresponding clean images $x$ (*upper-right*) by learning $p(x|y)$. It is clear that both MDM-dual and MDM-proj are capable of solving this conditional generation task, generating clean images that additionally embed invisible watermarks, *i.e.*, $x \in \mathcal{M}$.

is to generate clean, watermarked, images conditioned on degraded—restored with JPEG in this case—inputs. Similar to unconditional generation, we consider a polytope constraint set whose parameters are chosen such that the watermark yields high precision ($> 95\%$) and low false positive rate ($< 0.001\%$). Specifically, we set $m = 100$ and $b = -c = 1.2$. We initialize the networks with pretrained checkpoints from Liu et al. [75].

Figure 9 reports the qualitative results. It is clear that both MDM-dual and MDM-proj are capable of solving this conditional generation task, generating clean images that additionally embed invisible watermarks, *i.e.*, $x \in \mathcal{M}$. Note that all non-MDM-generated images, despite being indistinguishable, actually violate the polytope constraint, whereas MDM-generated images always satisfy the constraint. Overall, our results suggest that both MDM-dual and MDM-proj scale to high-dimensional applications and are capable of embedding invisible watermarks in high-resolution images.

## 6 Conclusion and Limitation

We developed Mirror Diffusion Models (MDMs), a dual-space diffusion model that enjoys simulation-free training for constrained generation. We showed that MDM outperforms prior methods in standard constrained generation, such as balls and simplices, and offers a competitive alternative in generating watermarked contents. It should be noted that MDM concerns mainly *convex* constrained sets, which, despite their extensive applications, limits the application of MDM to general constraints. It will be interesting to combine MDM with other dynamic generative models, such as flow-based approaches.

## Broader Impact

Mirror Diffusion Models (MDMs) advance the recent development of diffusion models to complex domains subjected to convex constrained sets. This opens up new possibilities for MDMs to serve as preferred models for generating samples that live in *e.g.,* simplices and balls. Additionally, MDMs introduce an innovative application of constrained sets as a watermarking technique. This has the potential to address concerns related to unethical usage and safeguard the copyright of generative models. By incorporating constrained sets into the generating process, MDMs offer a means to prevent unauthorized usage and ensure the integrity of generated content.

## Acknowledgements

The authors would like to thank Bogdon Vlahov for sharing the computational resources. GHL, TC, ET are supported by ARO Award #W911NF2010151 and DoD Basic Research Office Award HQ00342110002. MT is partially supported by NSF DMS-1847802, NSF ECCS-1942523, Cullen-Peck Scholarship, and GT-Emory AI.Humanity Award.

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

# A  Derivation of Mirror Mappings

Here, we provide addition derivation of $\nabla\phi^*$. Computation of $\nabla\phi(x)$ and $\nabla^2\phi^*(y)$ follow straightforwardly by differentiating $\phi(x)$ and $\nabla\phi^*(y)$ w.r.t. $x$ and $y$, respectively.

$\ell_2$-**Ball**  Since the gradient map also reverses the mirror map, we aim to rewrite $y = \frac{2\gamma}{R-\|x\|_2^2}x$ as $x = f(y) = \nabla\phi^*_{\text{ball}}(y)$. Solving the second-order polynomial,

$$\|y\|_2^2 = \left(\frac{2\gamma}{R-\|x\|_2^2}\right)^2 \|x\|_2^2, \tag{19}$$

yields

$$\|x\|_2^2 = R + \frac{2\gamma}{\|y\|_2^2}\left(\gamma - \sqrt{R\|y\|_2^2+\gamma^2}\right). \tag{20}$$

With that, we can rewrite Equation (11) by

$$x = \frac{R-\|x\|_2^2}{2\gamma}y \stackrel{(20)}{=} \frac{\sqrt{R\|y\|_2^2+\gamma^2}-\gamma}{\|y\|_2^2}y = \frac{R}{\sqrt{R\|y\|_2^2+\gamma^2}+\gamma}y.$$

**Simplex**  Standard calculations in convex analysis [28] shows

$$\phi^*_{\text{simplex}}(y) = \log\left(1+\sum_i^d e^{y_i}\right). \tag{21}$$

Differentiating Equation (21) w.r.t. $y$ yields $\nabla\phi^*_{\text{simplex}}$ in Equation (13).

**Polytope**  Since the gradient map also reverses the mirror map, we aim to inverse

$$y = \sum_{i=1}^m s_i(\langle a_i, x\rangle)a_i + \sum_{j=m+1}^d \langle a_j, x\rangle a_j. \tag{22}$$

When all $d$ constraints are orthonormal, taking inner product between $y$ and each $a$ yields

$$\langle a_i, y\rangle = s_i(\langle a_i, x\rangle), \qquad \langle a_j, y\rangle = \langle a_j, x\rangle. \tag{23}$$

Therefore, we can reconstruct $x$ from $y$ via

$$x = \sum_{i=1}^m \langle a_i, x\rangle a_i + \sum_{j=m+1}^d \langle a_j, x\rangle a_j$$

$$\stackrel{(23)}{=} \sum_{i=1}^m s_i^{-1}(\langle a_i, y\rangle)a_i + \sum_{j=m+1}^d \langle a_j, y\rangle a_j,$$

which defines $x = \nabla\phi^*_{\text{polytope}}(y)$. For completeness, the Hessian can be presented compactly as

$$\nabla^2\phi^*_{\text{polytope}}(y) = \boldsymbol{I} + \boldsymbol{A}\Sigma\boldsymbol{A}^\top, \tag{24}$$

where $\boldsymbol{I}$ is the identity matrix, $\boldsymbol{A} := [a_1,\cdots,a_m]$ is a $d$-by-$m$ matrix whose column vector $a_i$ corresponds to each constraint, and $\Sigma \in \mathbb{R}^{m\times m}$ is a diagonal matrix with leading entries

$$[\Sigma]_{ii} = \frac{\partial s_i^{-1}(z)}{\partial z}\Big|_{z=\langle a_i,y\rangle} - 1 \stackrel{(18)}{=} \frac{b_i-c_i}{2}\left(1-\tanh^2(\langle a_i,y\rangle)\right) - 1.$$

# B  Additional Remarks on Polytope

**Derivation of Equation (17)**  Since the subspaces spanned by $\{a_i\}$ and $\{a_j\}$ are orthogonal to each other, we can rewrite (15) as

$$\nabla\phi_{\text{polytope}}(x) = \sum_{i=1}^{m} s_i(\langle a_i, x\rangle)a_i + \left(x - \sum_{i=1}^{m}\langle a_i, x\rangle a_i\right) = x + \sum_{i=1}^{m}\left(s_i(\langle a_i, x\rangle) - \langle a_i, x\rangle\right)a_i.$$

$\nabla\phi^*_{\text{polytope}}(y)$ follows similar derivation.

**Generalization to non-orthonormal constraints**  The mirror maps of a polytope, as described in Equations (15) to (17), can be seen as operations that manipulate the coefficients associated with the bases defined by the constraints. This understanding allows us to extend the computation to non-orthonormal constraints by identifying the corresponding "coefficients" through a change of bases, utilizing the reproducing formula:

$$x = \sum_{i=1}^{d}\langle \tilde{a}_i, x\rangle a_i, \text{ where } \tilde{a}_i \text{ is the } i\text{-th row of } (\boldsymbol{A}^\top \boldsymbol{A})^{-1}\boldsymbol{A}^\top,$$

and $\boldsymbol{A} := [a_1, \cdots, a_m]$. Similarly, we have $y = \sum_{i=1}^{d}\langle\tilde{a}_i, y\rangle a_i$. Applying similar derivation leads to

$$\nabla\phi_{\text{poly}}(x) = x + \sum_{i=1}^{m}\left(s_i(\langle\tilde{a}_i, x\rangle) - \langle\tilde{a}_i, x\rangle\right)a_i, \ \nabla\phi^*_{\text{poly}}(y) = y + \sum_{i=1}^{m}\left(s_i^{-1}(\langle\tilde{a}_i, y\rangle) - \langle\tilde{a}_i, y\rangle\right)a_i.$$

# C  Experiment Details & Additional Results

Table 7: The concentration parameter $\alpha$ of each Dirichlet distribution in simplices constrained sets.

| $d$ | 3 | 3 | 7 | 9 | 20 |
|---|---|---|---|---|---|
| $\alpha$ | $[2, 4, 8]$ | $[1, 0.1, 5]$ | $[1, 2, 2, 4, 4, 8, 8]$ | $[1, 0.5, 2, 0.3, 0.6, 4, 8, 8, 2]$ | $[0.2, 0.4, \cdots, 4, 4.2]$ |

Table 8: Hyperparameters of the polytope $\mathcal{M} := \{x \in \mathbb{R}^d : c_i < a_i^\top x < b_i\}$ for each dataset and watermark precision. Note that we fix $b = b_i = -c_i$ in practice.

| | FFHQ 64×64 (uncon) | | | AFHQv2 64×64 (uncon) | | |
|---|---|---|---|---|---|---|
| *Precision* | *59.3%* | *71.8%* | *93.3%* | *56.9%* | *75.0%* | *92.7%* |
| Number of constraints $m$ | 7 | 20 | 100 | 4 | 20 | 100 |
| Constraint range $b$ | 1.05 | 1.05 | 1.05 | 0.9 | 0.9 | 0.9 |

**Dataset & constrained sets**

- $\ell_2$-*balls constrained sets*: For $d = 2$, we consider the Gaussian Mixture Model (with variance 0.05) and the Spiral shown respectively in Figures 1 and 3. For $d = \{6, 8, 20\}$, we place $d$ isotropic Gaussians, each with variance 0.05, at the corner of each dimension, and reject samples outside the constrained sets.

- *Simplices constrained sets*: We consider Dirichlet distributions [48], $\text{Dir}(\alpha)$, with various concentration parameters $\alpha$ detailed in Table 7.

- *Hypercube constrained sets*: For all dimensions $d = \{2, 3, 6, 8, 20\}$, we place $d$ isotropic Gaussians, each with variance 0.2, at the corner of each dimension, and either reject ($d = \{2, 3, 6, 8\}$) or reflect ($d = 20$) samples outside the constrained sets.

- *Watermarked datasets and polytope constrained sets*: We follow the same data preprocessing from EDM[5] [38] and rescale both FFHQ and AFHQv2 to 64×64 image resolution. For the polytope constrained sets $\mathcal{M} := \{x \in \mathbb{R}^d : c_i < a_i^\top x < b_i, \forall i\}$, we construct $a_i$ from orthonormalized Gaussian random vectors and detail other hyperparameters in Table 8.

---

[5]`https://github.com/NVlabs/edm`, released under Nvidia Source Code License.

**Implementation** All methods are implemented in PyTorch [76]. We adopt ADM[6] and EDM[5] [38] respectively as the MDM's diffusion backbones for constrained and watermarked generation. We implemented Reflected Diffusion [18] by ourselves as their codes have not yet been made available by the submission time (May 2023), and used the official implementation[7] of Reflected Diffusion [25] in Table 11. We also implemented Simplex Diffusion [63], but as observed in previous works [25], it encountered computational instability especially when computing the modified Bessel functions.

**Training** For constrained generation, all methods are trained with AdamW [77] and an exponential moving average with the decay rate of 0.99. As standard practices, we decay the learning rate by the decay rate 0.99 every 1000 steps. For watermarked generation, we follow the default hyperparameters from EDM[5] [38]. All experiments are conducted on two TITAN RTXs and one RTX 2080.

**Network** For constrained generation, all networks take $(y, t)$ as inputs and follow

$$\texttt{out} = \texttt{out\_mod}(\texttt{norm}(\texttt{y\_mod}(\ y\ ) + \texttt{t\_mod}(\texttt{timestep\_embedding}(\ t\ )))),$$

where `timestep_embedding(·)` is the standard sinusoidal embedding. `t_mod` and `out_mod` consist of 2 fully-connected layers (`Linear`) activated by the Sigmoid Linear Unit (`SiLU`) [78]:

$$\texttt{t\_mod} = \texttt{out\_mod} = \texttt{Linear} \to \texttt{SiLU} \to \texttt{Linear}$$

and `y_mod` consists of 3 residual blocks, *i.e.,* `y_mod`$(y) = y +$ `res_mod(norm`$(y)$`)`, where

$$\texttt{res\_mod} = \texttt{Linear} \to \texttt{SiLU} \to \texttt{Linear} \to \texttt{SiLU} \to \texttt{Linear} \to \texttt{SiLU} \to \texttt{Linear}$$

All `Linear`'s have 128 hidden dimension. We use group normalization [79] for all `norm`. For watermarked generation, we use EDM parameterization[5] [38].

**Evaluation** We compute the Wasserstein and Sliced Wasserstein distances using the `geomloss`[8] and `ot`[9] packages, respectively. The Maximum Mean Discrepancy (MMD) is based on the popular package https://github.com/ZongxianLee/MMD_Loss.Pytorch, which is unlicensed. For watermarked generation, we follow the same evaluation pipeline from EDM[5] [38] by first generating 50,000 watermarked samples and computing the FID w.r.t. the training statistics.

**False-positive rate** Similar to Kirchenbauer et al. [36], we reject the null hypothesis and detect the watermark if the sample produces no violation of the polytope constraint, *i.e.,* if $x \in \mathcal{M}$. Hence, the false-positive samples are those that are actually true null hypothesis (*i.e.,* not generated by MDM) yet accidentally fall into the constraint set, hence being mistakenly detected as watermarked. Specifically, the false-positive rates of our MDMs are respectively 0.07% and 0.08% for FFHQ and AFHQv2. Lastly, we note that the fact that both MDM-proj and MDM-dual generate samples that always satisfy the constraint readily implies 100% recall and 0% Type II error.

## C.1 Additional Results

**Tractable variational bound in Equation (8)** Figure 10 demonstrates how MDM faithfully captures the variational bound to the negative log-likelihood (NLL) of 2-dimensional GMM.

**More constrained sets, distributional metrics, & baseline** Tables 9 and 10 expand the analysis in Tables 3 and 4 with additional distributional metrics such as Wasserstein-1 ($\mathcal{W}_1$) and Maximum Mean Discrepancy (MMD). Additionally, Table 11 reports the results of hypercube $[0, 1]^d$ constrained set, a special instance of polytopes, and includes additional baseline from Lou and Ermon [25], which approximate the

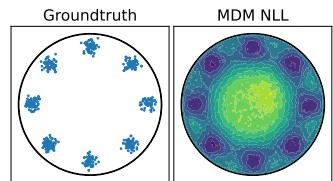

Figure 10: Tractable variational bound by our MDM.

---

[6]https://github.com/openai/guided-diffusion, released under MIT License.
[7]https://github.com/louaaron/Reflected-Diffusion, latest commit (65d05c6) at submission, unlicensed.
[8]https://github.com/jeanfeydy/geomloss, released under MIT License.
[9]https://pythonot.github.io/gen_modules/ot.sliced.html#ot.sliced.sliced_wasserstein_distance, released under MIT License.

intractable scores in Reflected Diffusion using eigenfunctions tailored specifically to hypercubes, rather than implicit score matching as in Fishman et al. [18]. Consistently, our findings conclude that the MDM is the *only* constrained-based diffusion model that achieves comparable or better performance to DDPM. These results affirm the effectiveness of MDM in generating high-quality samples within constrained settings, making it a reliable choice for constrained generative modeling.

**More watermarked samples** Figures 11 and 12 provide additional qualitative results on the watermarked samples generated by MDMs.

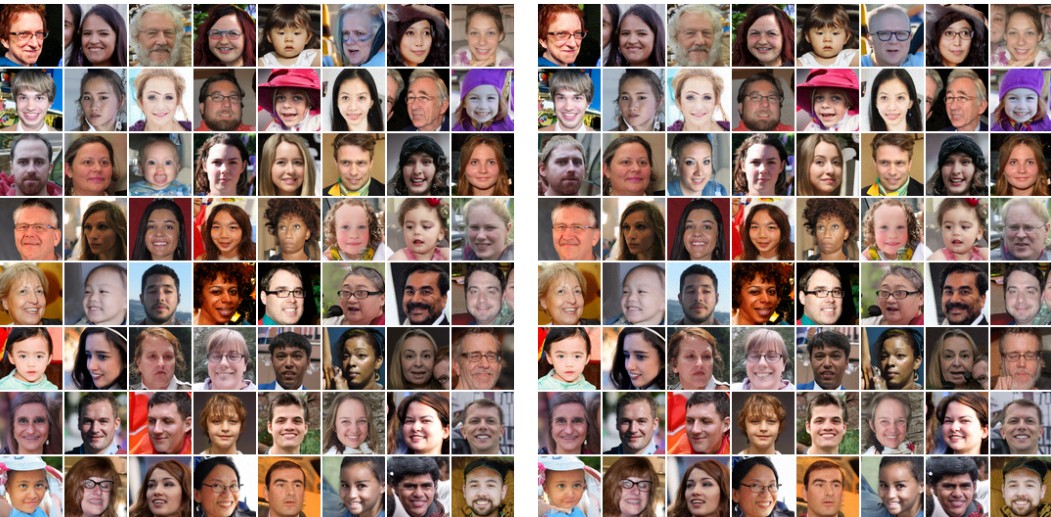

Figure 11: FFHQ 64×64 unconditional watermarked samples generated by (**left**) MDM-proj and (**right**) MDM-dual from the same set of random seeds. Despite the fact that some images, such as the one in the first row and sixth column, were altered possibly due to the change of dual-space distribution (see Figure 7), they look realistic and remain close to the data distribution.

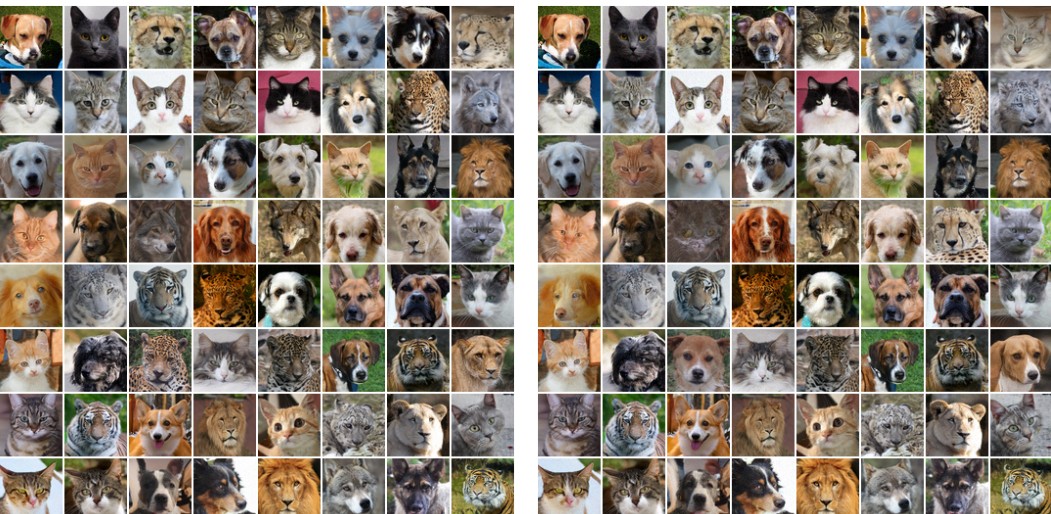

Figure 12: AFHQv2 64×64 unconditional watermarked samples generated by (**left**) MDM-proj and (**right**) MDM-dual from the same set of random seeds. Despite the fact that some images, such as the one in the fifth row and first column, were altered possibly due to the change of dual-space distribution (see Figure 7), they all look realistic and remain close to the data distribution.

Table 9: Expanded results of $\ell_2$-**ball constrained sets**, where we include additional distributional metrics such as $\mathcal{W}_1$ and Maximum Mean Discrepancy (MMD), all computed with 1000 samples and averaged over three trials. Consistently, our findings conclude that the MDM is the *only* constrained-based diffusion model that achieves comparable or better performance to DDPM.

| | $d=2$ | $d=2$ | $d=6$ | $d=8$ | $d=20$ |
|---|---|---|---|---|---|
| $\mathcal{W}_1 \downarrow$ (unit: $10^{-2}$) | | | | | |
| DDPM [2] | $0.66 \pm 0.15$ | $0.14 \pm 0.03$ | $\mathbf{0.52} \pm 0.09$ | $\mathbf{0.58} \pm 0.10$ | $3.45 \pm 0.50$ |
| Reflected [18] | $0.55 \pm 0.29$ | $0.46 \pm 0.17$ | $3.11 \pm 0.40$ | $10.13 \pm 0.21$ | $19.42 \pm 0.13$ |
| MDM (ours) | $\mathbf{0.46} \pm 0.07$ | $\mathbf{0.12} \pm 0.04$ | $0.72 \pm 0.39$ | $1.05 \pm 0.26$ | $\mathbf{2.63} \pm 0.31$ |
| MMD $\downarrow$ (unit: $10^{-2}$) | | | | | |
| DDPM [2] | $0.67 \pm 0.23$ | $\mathbf{0.23} \pm 0.07$ | $\mathbf{0.37} \pm 0.19$ | $0.75 \pm 0.24$ | $0.98 \pm 0.42$ |
| Reflected [18] | $0.58 \pm 0.46$ | $5.03 \pm 1.17$ | $2.34 \pm 0.14$ | $28.82 \pm 0.66$ | $14.83 \pm 0.62$ |
| MDM (ours) | $\mathbf{0.52} \pm 0.36$ | $0.27 \pm 0.19$ | $0.54 \pm 0.12$ | $\mathbf{0.35} \pm 0.23$ | $\mathbf{0.50} \pm 0.17$ |
| Constraint violation (%) $\downarrow$ | | | | | |
| DDPM [2] | $0.00 \pm 0.00$ | $0.00 \pm 0.00$ | $8.67 \pm 0.87$ | $13.60 \pm 0.62$ | $19.33 \pm 1.29$ |

Table 10: Expanded results of **simplices constrained sets**.

| | $d=3$ | $d=3$ | $d=7$ | $d=9$ | $d=20$ |
|---|---|---|---|---|---|
| $\mathcal{W}_1 \downarrow$ (unit: $10^{-2}$) | | | | | |
| DDPM [2] | $\mathbf{0.01} \pm 0.00$ | $0.02 \pm 0.01$ | $\mathbf{0.03} \pm 0.00$ | $\mathbf{0.05} \pm 0.00$ | $\mathbf{0.11} \pm 0.00$ |
| Reflected [18] | $0.06 \pm 0.01$ | $0.12 \pm 0.00$ | $0.62 \pm 0.08$ | $3.57 \pm 0.05$ | $0.98 \pm 0.02$ |
| MDM (ours) | $\mathbf{0.01} \pm 0.00$ | $\mathbf{0.01} \pm 0.01$ | $\mathbf{0.03} \pm 0.00$ | $\mathbf{0.05} \pm 0.00$ | $0.13 \pm 0.00$ |
| MMD $\downarrow$ (unit: $10^{-2}$) | | | | | |
| DDPM [2] | $0.72 \pm 0.07$ | $0.72 \pm 0.30$ | $0.74 \pm 0.10$ | $0.97 \pm 0.22$ | $1.12 \pm 0.07$ |
| Reflected [18] | $3.91 \pm 0.95$ | $15.12 \pm 1.36$ | $16.48 \pm 1.04$ | $131.44 \pm 2.65$ | $57.90 \pm 2.07$ |
| MDM (ours) | $\mathbf{0.44} \pm 0.16$ | $\mathbf{0.50} \pm 0.26$ | $\mathbf{0.42} \pm 0.08$ | $\mathbf{0.55} \pm 0.13$ | $\mathbf{0.61} \pm 0.03$ |
| Constraint violation (%) $\downarrow$ | | | | | |
| DDPM [2] | $0.73 \pm 0.12$ | $14.40 \pm 1.39$ | $11.63 \pm 0.90$ | $27.53 \pm 0.57$ | $68.83 \pm 1.66$ |

Table 11: Results of **hypercube** $[0, 1]^d$ **constrained sets**.

| | $d=2$ | $d=3$ | $d=6$ | $d=8$ | $d=20$ |
|---|---|---|---|---|---|
| Sliced Wasserstein $\downarrow$ (unit: $10^{-2}$) | | | | | |
| DDPM [2] | $\mathbf{2.24} \pm 1.22$ | $2.17 \pm 0.65$ | $2.05 \pm 0.41$ | $2.01 \pm 0.16$ | $\mathbf{1.54} \pm 0.01$ |
| Reflected [25] | $3.75 \pm 1.20$ | $6.58 \pm 1.18$ | $2.77 \pm 0.06$ | $3.50 \pm 0.69$ | $3.37 \pm 0.46$ |
| Reflected [18] | $19.05 \pm 1.51$ | $17.16 \pm 0.88$ | $11.90 \pm 0.43$ | $7.49 \pm 0.13$ | $4.32 \pm 0.23$ |
| MDM (ours) | $3.00 \pm 0.72$ | $\mathbf{1.92} \pm 0.81$ | $\mathbf{1.75} \pm 0.17$ | $\mathbf{1.85} \pm 0.34$ | $3.35 \pm 0.64$ |
| $\mathcal{W}_1 \downarrow$ (unit: $10^{-2}$) | | | | | |
| DDPM [2] | $\mathbf{0.07} \pm 0.05$ | $0.22 \pm 0.07$ | $1.65 \pm 0.14$ | $\mathbf{3.30} \pm 0.16$ | $\mathbf{16.74} \pm 0.12$ |
| Reflected [25] | $0.20 \pm 0.12$ | $1.21 \pm 0.39$ | $2.53 \pm 0.04$ | $4.82 \pm 0.42$ | $25.47 \pm 0.20$ |
| Reflected [18] | $4.40 \pm 0.57$ | $6.01 \pm 0.97$ | $9.34 \pm 0.56$ | $9.84 \pm 0.24$ | $25.27 \pm 0.36$ |
| MDM (ours) | $0.08 \pm 0.03$ | $\mathbf{0.20} \pm 0.07$ | $\mathbf{1.57} \pm 0.08$ | $3.34 \pm 0.23$ | $20.59 \pm 1.19$ |
| MMD $\downarrow$ (unit: $10^{-2}$) | | | | | |
| DDPM [2] | $0.27 \pm 0.26$ | $0.32 \pm 0.14$ | $0.69 \pm 0.21$ | $0.81 \pm 0.23$ | $0.73 \pm 0.07$ |
| Reflected [25] | $0.92 \pm 0.53$ | $3.56 \pm 1.31$ | $1.16 \pm 0.04$ | $2.09 \pm 0.70$ | $2.83 \pm 0.58$ |
| Reflected [18] | $32.26 \pm 3.19$ | $26.64 \pm 5.07$ | $29.83 \pm 1.42$ | $15.84 \pm 0.89$ | $7.21 \pm 0.68$ |
| MDM (ours) | $\mathbf{0.27} \pm 0.09$ | $\mathbf{0.29} \pm 0.17$ | $\mathbf{0.39} \pm 0.14$ | $\mathbf{0.61} \pm 0.23$ | $\mathbf{0.62} \pm 0.05$ |
| Constraint violation (%) $\downarrow$ | | | | | |
| DDPM [2] | $9.37 \pm 0.12$ | $17.57 \pm 1.27$ | $41.70 \pm 1.30$ | $59.30 \pm 1.39$ | $94.47 \pm 0.64$ |

