# OpenReview forum: "Mirror Diffusion Models for Constrained and Watermarked Generation"
_NeurIPS.cc/2023/Conference — NeurIPS 2023 poster_

### Official Review · Reviewer_VdTP · 2023-07-02

**Soundness:** 4 excellent
**Presentation:** 3 good
**Contribution:** 3 good
**Rating:** 6
**Confidence:** 3

**Summary:**

This paper proposes a new class of diffusion models called the Mirror Diffusion Model (MDM), which confines the generation to a constrained convex set. The MDM transforms the generation from a constrained original space to an unconstrained dual space. With this transformation, MDM can be trained and sampled like the unconstrained Euclidean-space diffusion models, such as DDPM, with an additional step mapping back to the original space. In experiments, the authors show that the general quality of MDM on constrained sets outperforms the reflected diffusion models in terms of quality and efficiency. Furthermore, the authors demonstrate an important application of MDM in watermark generation.

**Strengths:**

(S1) This paper proposes a new class of diffusion models that confine the generation to a constrained convex set. It achieves this by transforming the generation from the constrained original space to an unconstrained dual space, enabling efficient training and sampling similar to unconstrained diffusion models.

(S2) The mechanism relies on a strongly convex function defined on the convex set. This function's gradient space spans $R^d$, and the gradient norm approaches infinity near the boundary of the convex set. The author provides clear instructions on how to design such a convex function for various shapes, including the $\ell_2$-ball, simplex, and general polytopes.

(S3) The experiments demonstrate that the proposed model outperforms reflected diffusion models in terms of generation quality and efficiency. It also achieves generation quality comparable to that of unconstrained DDPM without violating any constraints.

**Weaknesses:**

(W1) From my understanding, the proposed model requires the design of a strongly convex function whose gradient maps the constrained convex set to the entire $R^d$ space, enabling the application of an unconstrained diffusion model in the dual space. However, it is not immediately apparent to me how the authors ensure that this condition holds for the strongly convex functions they provide.

(W2) It would indeed be beneficial to observe experiments conducted on more realistic datasets, such as CIFAR-100 and ImageNet.

**Questions:**

(Q1) Can the authors provide the proofs that gradient of the strongly convex function maps the constrained convex set to the whole $R^d$?

(Q2) Can the authors provide more experimental results on more realistic data, like CIFAR-100 and ImageNet?

---

> ### Author Rebuttal · Authors · 2023-08-09
>
> **1. Conditions when $\nabla\phi(\mathcal{M})= \mathbb{R}^d$**
>
> - We first note that the gradient map of a strictly convex function $\phi$ needs *not* span $\mathbb{R}^d$, unless additional conditions are satisfied. For mirror maps, we follow the literature (e.g., [1,2]) and require $\phi$ to be additionally *(i)* of Legendre type [1] (i.e., $\lim_{x\rightarrow\partial\mathcal{M}}\|\nabla\phi(x)\|\rightarrow \infty$, see L118 Sec 3.1) and *(ii)* continuous differentiable (see L117 Sec 3.1). When $\phi$ also satisfies these two conditions, its gradient map will be surjective with range $\mathbb{R}^d$.
>
> - The above surjectivity statement follows from convex analysis (e.g., [1]). Here is how we understood it intuitively. For gaining insight consider a 1D case. Since $\phi$ is strictly convex, its derivative is strictly increasing (see, e.g., [1,3]). Then, the above two conditions ensure that the derivative not only approach $(-\infty,\infty)$ at the boundary $\partial\mathcal{M}$, but is also continuous in its domain (i.e., there exists no hole or jump). Hence, the gradient map has range $\mathbb{R}$.
>
> - It is, however, a valid and in fact critical question, whether such requirements can be satisfied for any convex constraint set (i.e. the mirror map approach works). Unfortunately, to the best of our knowledge, how to explicitly construct a mirror map (with both $\nabla \phi$ and $(\nabla \phi)^{-1}$ analytically given) for an arbitrary convex set is still an open problem, although existence is less an issue.
>
> ---
>
> **2. Large-scale experiments on watermarking ImageNet 256x256**
> - We appreciate the reviewer's comment. In the **PDF attached above in our response to all reviewers**, we present additional results of our MDMs on large-scale image datasets, specifically ImageNet 256x256, for both conditional and unconditional watermarked generation. For conditional generation, we focus on image restoration tasks, generating clean, watermarked, images conditioned on degraded inputs, using both MDM-dual and MDM-proj. For unconditional generation, we include mainly MDM-proj due to time constraints during rebuttal. Both MDM-proj and MDM-dual consider a polytope constraint set whose parameters are chosen such that the watermark yields high precision (>95%) and low false positive rate (< 0.001%). Specifically, we set $m$=100, $b$=1.2, $c$=-1.2, and $a_i \in \mathbb{R}^{196608}$ orthogonal Gaussian random vectors. Similar to Sec 5.2, we initialize networks with pretrained checkpoints [4,5].
>
> - Our qualitative results suggest that both MDM-dual and MDM-proj scale to high-dimensional applications and are capable of embedding invisible watermarks in high-resolution images. Note that all non-MDM-generated images, despite being indistinguishable, actually violate the polytope constraint, whereas MDM-generated images always satisfy the constraint. These results highlight the scalability of our MDM for large-scale, high-dimensional applications. We will release our codes upon publication.
>
> ---
>
> [1] Convex analysis. (Rockafellar 1970)
> [2] The Mirror Langevin Algorithm Converges with Vanishing Bias
> [3] https://math.stackexchange.com/questions/999550/strictly-convex-if-and-only-if-derivative-strictly-increasing
> [4] https://github.com/openai/guided-diffusion
> [5] https://github.com/NVlabs/I2SB

---

> > ### Comment · Reviewer_VdTP · 2023-08-14
> >
> > I appreciate the authors' comprehensive explanations. I will certainly factor these points into my deliberations during the discussion with the AC.

---

### Official Review · Reviewer_Bc59 · 2023-07-06

**Soundness:** 3 good
**Presentation:** 4 excellent
**Contribution:** 3 good
**Rating:** 6
**Confidence:** 4

**Summary:**

When the data distribution is constrained in some boundary,
This paper introduced Mirror Diffusion Models (MDM), where the diffusion process runs not in the distribution of the (constrained) primal space, but in the distribution of the (unconstrained) dual space. For constrained datasets such as simplex, polytopes, and balls. While the primal space is kept constrained, the dual space is unconstrained and the standard Gaussian diffusion process can be used without any restrictions. When adequate dual functions and mirror maps are given, this paper achieved better sampling quality when generating from a constrained set, both in small-dimensional synthetic datasets and real-world datasets. And this offers competitive results in generating watermarked images.

**Strengths:**

* The paper has good clarity in introducing theorems, examples, and applications.
* To the best of our knowledge, this is the first work that necessitates and uses a primal-dual algorithm in the diffusion model literature. Given well-defined dual functions and mirror maps, the unconstrained diffusion model can be used for generating constrained datasets. The experiments with synthetic datasets (Dirichlet distribution) provides evidence to use MDM in categorical distribution, and competitive results in watermarked distribution provides further direction in privacy issues in diffusion-based generative models.

**Weaknesses:**

* The performance gap between MDM-proj and MDM-dual is not narrowed. When the U-Net-based network for images are specialized in generating pixel-based images, the dual-space distribution can be less specialized to be sampled using the same network architectures.

**Questions:**

* How many sampling steps did you used for generating in each dataset?

=====

Correction
* One of the xlabels in Figure 6 should be `MDM-dual`. Now both two are `MDM-proj`.
* Line 514: polytop --> polytope
* Figure 8: top and bottom --> left and right

**Limitations:**

The authors adequately addressed the limitations.

---

> ### Author Rebuttal · Authors · 2023-08-09
>
> **1. Performance of dual-space diffusion models**
>
> - We first thank the reviewer for raising the comment. While we do notice a gap between `MDM-proj` and `MDM-dual` (first 2 rows in Table 6), we stress that these FID values are evaluated w.r.t. the original, *constraint-violated*, training set distribution, which differs from the dual-space, *constraint-satisfied*, distribution from which `MDM-dual` was trained. An alternative, arguably more suitable, metric for evaluating `MDM-dual` is the FID w.r.t. the dual-space distribution, which we include in the last row of Table 6 (highlighted in gray). There, `MDM-dual` behaves similarly to `MDM-proj`, exhibiting a FID-Precision trade-off. Qualitatively, Fig 6,10,11 show that `MDM-dual` is able to generate watermarked images with good quality.
>
> - Nevertheless, whether the current parametrization, using U-net, best suits learning dual-space diffusion models is an interesting question, and we believe it will depend strongly on the choice of (polytope) constraint sets, since dual-space samples essentially change the coefficient bases, defined by the constraint set, of primal-space samples (see Eq 17). For image applications, we find that MDM can best embed invisible watermarks using high frequency $a_i$, as it preserves the semantic structure of images. Co-designing parametrization with the constraint set will be an interesting future direction to pursue, and we thank the reviewer for raising the comment.
>
> ---
>
> **2. Typo & other clarifications**
>
> - We thank the reviewer for the meticulous reading. There’s indeed a typo in Fig 6. The right 2 columns should be `MDM-dual` rather than `MDM-proj`. All noted typos will be fixed in the revision (kindly note that we are not allowed to revise the submission at the moment).
>
> - Regarding sampling steps, all datasets and diffusion models in Sec 5.1 generate samples with 1000 steps, as mentioned in L239. For image datasets, FFHQ and AFHQv2, in Sec 5.2, we generate watermarked images with 79 steps, following the setup from EDM [1]. We note, however, that we did not perform extensive tuning on this hyper-parameter.
>
> ---
>
> [1] Elucidating the design space of diffusion models

---

> > ### Comment · Reviewer_Bc59 · 2023-08-13
> > **Response to the official review**
> >
> > Thank you to the authors for the thoughtful responses. I keep the current score.

---

### Official Review · Reviewer_kFs5 · 2023-07-07

**Soundness:** 3 good
**Presentation:** 3 good
**Contribution:** 3 good
**Rating:** 6
**Confidence:** 4

**Summary:**

This paper studies how to learning diffusion model when the data is in a constrained domain. The idea is to map the data into an unconstrained domain using the mirror map and conduct the diffusion on that mirror space. Once the generation finishes, map the data back to the original space.

**Strengths:**

Using the idea of mirror map seems a novel approach.
This approach also allows for likelihood computation, which is helpful for model evaluation.

**Weaknesses:**

The method seems to be limited to several types of special domains. What's the challenge of generalizing it into more general domain constraints?

The experiment session can be improved. For example, to evaluate the model's performance on generating the data on simplex, we can consider generating the segmentation maps. Some larger scale experiment would be helpful.

Some very related literature on learning diffusion models on constrained domains are missing [1, 2] and need to be discussed.

[1] Learning Diffusion Bridges on Constrained Domains
[2] First Hitting Diffusion Models for Generating Manifold, Graph and Categorical Data

**Questions:**

See above.

**Limitations:**

See above.

---

> ### Author Rebuttal · Authors · 2023-08-09
>
> **1. Discussion on [1,2]**
>
> - We first thank the reviewer for bringing up these missing references, which are indeed relevant to our Mirror Diffusion Model (MDM). Both [1,2] and our MDM generate samples in constrained domains. While [1,2] can be applied to, e.g., discrete domains [1,2], equality constraints (e.g., sphere boundary) [2], and products of 1D bounded intervals [1], our MDMs instead focus on “**constrained sets”**, i.e., a particular class of constrained domains that are specified by “**inequality constraints**”, $\mathcal{M}:=${$x\in\mathbb{R}^d: f_i(x)<0, \forall i $}. Hence, MDM stands in parallel to [1,2]. Similar to [1,2], MDM also enjoys simulation-free training and adopts regression objectives (see Table 1), making MDM superior to prior inequality-constrained diffusion models [3,4], as evidenced by Sec 5.1 (Table 3,4,5, Fig 5) and Appendix C.1 (Table 9,10,11). Finally, we note that our MDM is the first to explore inequality-constrained domains as a new mechanism for watermarked generation, which is otherwise absent in [1,2,3,4].
>
> - As we do acknowledge that [1,2] are important references, we will include them, along with the above discussions, in the following revision (kindly note that we are not allowed to revise the submission at the moment).
>
> ---
>
> **2. Generalization to other constrained domains**
>
> - Following the discussion in **1.**, we re-emphasize that the goal of our MDM is to generate samples confined to “**inequality constraints**”. In principle, MDM can generate samples confined to *any* convex constraint set given its mirror map. In Sec 3.2, we exemplify three mirror maps, each for a different type of constraint set, mainly to demonstrate how efficient, closed-form mirror maps can be constructed for most inequality constraints considered in prior works [3,4]. This, as also recognized by Reviewer i5Cc, can be beneficial to a broader audience. For general convex constraint sets, MDM still remains applicable by constructing, e.g., log-barriers. While this may induce additional cost at inference, it introduces *no* computational overhead at training, which, crucially, preserves all desired computational advantages from Euclidean-space diffusion models (see Table 1).
>
> - As mirror maps are, by construction, built from convex functions, our MDM is also subjected to their domains (see L283-284 in Sec 6). We note, however, that MDM may still be applicable to general (non-convex), yet compact, constraint sets by adopting the diffeomorphism discussed in [4] (see their Fig 2 (iv)). Constructing simulation-free diffusions (like our MDM) for more general inequality constraints is an interesting future direction worth pursuing, and we thank the reviewer for raising these comments.
>
> ---
>
> **3. Large-scale experiments on watermarking ImageNet 256x256**
>
> - We appreciate the reviewer's comment. In the **PDF attached above in our response to all reviewers**, we present additional results of our MDMs on large-scale image datasets, specifically ImageNet 256x256, for both conditional and unconditional watermarked generation. For conditional generation, we focus on image restoration tasks, generating clean, watermarked, images conditioned on degraded inputs, using both MDM-dual and MDM-proj. For unconditional generation, we include mainly MDM-proj due to time constraints during rebuttal. Both MDM-proj and MDM-dual consider a polytope constraint set whose parameters are chosen such that the watermark yields high precision (>95%) and low false positive rate (< 0.001%). Specifically, we set $m$=100, $b$=1.2, $c$=-1.2, and $a_i \in \mathbb{R}^{196608}$ orthogonal Gaussian random vectors. Similar to Sec 5.2, we initialize networks with pretrained checkpoints [5,6].
>
> - Our qualitative results suggest that both MDM-dual and MDM-proj scale to high-dimensional applications and are capable of embedding invisible watermarks in high-resolution images. Note that all non-MDM-generated images, despite being indistinguishable, actually violate the polytope constraint, whereas MDM-generated images always satisfy the constraint. These results highlight the scalability of our MDM for large-scale, high-dimensional applications. We will release our codes upon publication.
>
> ---
>
> [3] Diffusion Models for Constrained Domains
> [4] Reflected Diffusion Models
> [5] https://github.com/openai/guided-diffusion
> [6] https://github.com/NVlabs/I2SB

---

> > ### Comment · Reviewer_kFs5 · 2023-08-18
> > **Thanks**
> >
> > The rebuttal partially addresses my concerns and I thus increase my score.

---

### Official Review · Reviewer_i5Cc · 2023-07-10

**Soundness:** 4 excellent
**Presentation:** 2 fair
**Contribution:** 3 good
**Rating:** 7
**Confidence:** 4

**Summary:**

The submission "Mirror Diffusion Models for Constrained and Watermarked Generation" describes a new approach to generate constrained data with diffusion models. Using mirror maps, the diffusion process proceeds as usual in an unconstrained space, but the generated data can be converted into the constraint space through the bijective mirror map.

The submission discusses (toy) applications in constrained generation onto a ball and a simplex, and an application of this approach to watermarking of diffusion model outputs.

**Strengths:**

The proposed approach using tools from convex optimization to encode convex constraints into the generation process is simple and elegant. The submission also spends a good amount of time on exposition of examples for commonly-used mirror maps, which I think will be very beneficial to the wider readership.
The application to watermarking is a suprising, but interesting connection that the submission draws, discussing an immediate beneficial application of the proposed approach. Overall I consider this a strong submission.

**Weaknesses:**

I think I fully understood the submission up to Section 5.2, which is quite compressed (probably due to space reasons), and I found it hard to fully understand. I'll enumerate these questions below in the questions section.


Otherwise, I see no major weaknesses. There are some typos, which I'll briefly mention:
* In standard Euclidean spaces, tractable marginal can be
* The marginal q(yt−1 |yt , y0 ) hints the optimal reverse

Appendix:
* Boarder impact
*  polytop

**Questions:**

* Just for my understanding: For MDM-proj, the output of a given, pretrained diffusion model, is project onto the constraint set after generation? And for MDM-dual, a new model is trained/fine-tuned is trained on contraint-projected samples?
* The polytope constraint for the watermark is hard for me to understand intuitively, what is being constrained here? The pixel space of the generated image is constrained to fulfil a random projection into a random interval? Intuitively, it is not clear to me why the impact of this constraint on the generated images is not larger?
* Related to question 1, does this mean that only MDM-proj can use multiple tokens for multiple users, and MDM-dual needs to retrain a new model for every new token?
* Why is the precision of the watermark less than 100%? From my understanding of the preceding sections, a constraint violation should be excluded? Why are images being generated that violate the constraints?

Overall, I hope the authors could clarify these questions and rewrite Section 5.2 to be easier to understand.

A few more questions/comments:

* The functions phi are almost Legendre functions in the sense that they are essentially smooth and essentially strictly convex. Is the twice-differentiability needed in principle (aside from Eq.(8)?
* The reduction to a bjiective map in Eq.(18) is a bit unsatisfying. I think it would be neater to derive the base function for this tanh shift
* Slightly related, Legendre functions do not seem to be strictly needed to generate the mirror maps in this work? Technically, a strictly monotone operator (as defined e.g. in "Convex Analysis and Monotone Operator Theory in Hilbert Spaces" ) would be sufficient? (This is more of a side question concerning the notation, I don't think that too much is gained by generalizing this way)
* Is it possible to derive p-values for the watermarked model to quantify the uncertainty about the watermark, as was done in Kirchenbauer 2023? From a practical perspective watermark, precision is not the primary objective for a watermark, but precision with a low false-positive rate of the resulting detection scheme.

**Limitations:**

The authors sufficiently discuss broader impact in the appendix.

---

> ### Author Rebuttal · Authors · 2023-08-09
>
> **1. Clarification on watermark generation in Sec 5.2**
>
> - The reviewer’s understanding of MDM-proj and MDM-dual is correct: MDM-proj projects samples generated by pretrained diffusion models to a constraint set whose parameters (i.e., tokens) are visible only to the private user. In contrast, MDM-dual learns a dual-space diffusion model from the constrained-projected samples. Hence, as conjectured by the reviewer, MDM-proj allows multiple tokens for multiple users, whereas MDM-dual, like other MDMs in Sec 5.1, is constraint-dependent.
> - We view images as samples in the vectorized image space $\mathbb{R}^d$. To watermark a, e.g., 64x64 image ($d$=3x64x64=12288), we consider a polytope constraint set, $\mathcal{M}:=${$x\in \mathbb{R}^d: c_i < a_i^\top x < b_i, \forall i \in [m]$}, where $a_i \in \mathbb{R}^d$ are orthonormal vectors in the **vectorized image space**. Hence, projecting $x$ to $\mathcal{M}$ constrains the signed distances of $x$ to these linear-independent hyperplanes (defined by $a_i$’s) to be within the intervals $(c_i, b_i)$.
> - The impact of the polytope constraint on generation quality depends on the choice of {$a_i, b_i, c_i$}$_{i=1}^m$, as illustrated by the ablation study in Fig 8. For image applications, we find that watermarks can be invisibly embedded with high-frequency $a_i$’s, larger interval of $c_i$’s and $b_i$’s, and a larger $m$. This is because high-frequency perturbations often preserve the semantic structure of images. While a larger interval improves the generation quality at the cost of loosening the constraint set, the overall precision is tightened up by increasing the number of constraints, $m$.
> - Similar to Kirchenbauer 2023, we reject the null hypothesis and detect the watermark if the sample produces no violation of the polytope constraint, i.e., if $x \in \mathcal{M}$. The reviewer is correct that both MDM-proj and MDM-dual generate samples that always satisfy the constraint. This readily implies 100% recall (`TP/(TP+FN)`) and 0% Type II error (`FN`), yet *not* necessarily 100% precision (`TP/(TP+FP)`) due to false-positive (`FP`) samples. Specifically, `FP` samples are those that are actually true null hypothesis (i.e., *not* generated by MDM) yet accidentally fall into the constraint set, hence being mistakenly detected as watermarked.
> (Notation: `TP`,`FP`,`TN`,`FN` respectively denote the numbers of True Positives, False Positives, True Negatives, and False Negatives.)
> - In the table below, we report the precision, false-positive rate (FPR), and accuracy of our MDMs that are used to generate all image figures (Fig 1,6,10,11). We stress that on both datasets, our MDMs achieve high accuracy & precision with low FPR.
>
>     |  | Precision (`TP/(TP+FP)`) | FPR (`FP/(FP+TN)`) | Accuracy (`(TP+TN)/(TP+FP+TN+FN)`) |
>     | --- | --- | --- | --- |
>     | FFHQ | 93.3% | 0.072% | 96.4% |
>     | AFHQv2 | 92.7% | 0.079% | 96.1% |
>
> - We do acknowledge that Sec 5.2 could be stated clearly with additional paragraphs, yet we were limited by the space constraint at the submission time. Admitted that we are still not allowed to revise the submission, we will rewrite Sec 5.2 and include these discussions in the later revision. We thank the reviewer again for raising these comments.
>
> ---
>
> **2. Relation of $\phi$ to Legendre function**
> - We first thank the reviewer for raising this interesting comment. We based our notation (in Sec 3) on the literature of Mirror Langevin Dynamics (MLD), e.g., [1]. There $\phi$ was indeed required to be of “Legendre type [2]” and twice-differentiable. This was because the twice differentiability was used to induce a (Riemannian) metric on the mirror space, from which MLD is constructed. For our MDM, however, twice differentiability is indeed needed only for Eq 8 as the reviewer pointed out. Continuous differentiability, i.e., $C^1$, would suffice for training. We very much appreciate this comment and will clarify this detail in the revision.
> - In this view, we may indeed be able to generalize our approach to strictly monotone operators. Meanwhile, one handy feature of using a convex $\phi$ is, the inverse of its Hessian (inverse in the sense of matrix inverse) is the Hessian of its dual. Algorithmically, this might not be necessarily, but how it may affect the performance is not yet clear to us. Therefore, this generalization, despite beyond the scope of our submission, will be an interesting future direction, and we also think other applications such as constrained generation in function spaces would be very interesting. Thank you for asking!
>
> ---
>
> **3. Typo & other clarifications**
> - All noted typos will be fixed in the revision. Additionally, we will rewrite Eq 18, the tanh shift, in the form of base functions. We thank the reviewer for the meticulous reading!
>
> ---
>
> [1] The Mirror Langevin Algorithm Converges with Vanishing Bias
> [2] Convex Analysis (Rockafellar 1970)

---

> > ### Comment · Reviewer_i5Cc · 2023-08-12
> > **Thanks**
> >
> > Thank you for providing this detailed response, it would be great if these clarifications could be included in the update. I have no further questions!

---

### Author Rebuttal · Authors · 2023-08-09

### **Author response to all reviewers**

We thank the reviewers for their valuable comments. We are excited that the reviewers identified the novelty of using mirror maps in learning constrained diffusion models ( **Reviewers** ****i5Cc****, ****kFs5, Bc59****, ****VdTP****), acknowledged our superior empirical results over prior diffusion models [1,2] and distinct application to watermarked generation (**Reviewers** ****i5Cc****, ****Bc59****, ****VdTP****), and found the paper well-written (**Reviewers** ****kFs5****, ****Bc59****, ****VdTP****). We believe MDM takes a significant step toward a new class of tractable diffusion models for constrained sets and watermarked generation.

As *all* reviewers recognized our technical contribution, one of the criticisms stemmed from the insufficient evaluation of our MDM on larger-scale datasets (raised by Reviewers kFs5, VdTP). In the **attached PDF (below)**, we present additional results of our MDMs, on ImageNet 256x256, for both conditional and unconditional watermarked generation. Our qualitative results suggest that MDMs can be scaled to high-dimensional application, and are capable of embedding invisible watermarks in high-resolution images. While all non-MDM-generated images, despite being indistinguishable, violate the constraint/token, our MDM satisfies the constraint by construction. We highlight these new results, gained uniquely in MDMs by constructing efficient mirror maps, that are otherwise absent in prior diffusion methods [1,2].

We tried our best to resolve all raised questions in the individual responses below. If you have any additional questions/comments/concerns, please let us know. We always appreciate the reviewer's precious time in providing their valuable feedback.

[1] Diffusion Models for Constrained Domains
[2] Reflected Diffusion Models

---

**Please find additional figures in the PDF below**:

---

### Decision · Program_Chairs · 2023-09-21

**Decision:**

Accept (poster)

**Comment:**

In this paper, the authors studied diffusion generative modeling when the data are confined to some constrained set, which goes beyond the standard Euclidean space. A new mechanism called the mirror diffusion model (inspired by mirror descent in the optimization literature) has been proposed, which employs suitable mirror maps to exploit the structure of various constrained sets. Intriguing performance has been shown for the application of watermarking. The paper would benefit from more experiments on larger-scale real-world datasets.